# Evaluation of the Analgesic Efficacy of a Bioelectronic Device in Non-Specific Chronic Low Back Pain with Neuropathic Component. A Randomized Trial

**DOI:** 10.3390/jcm10081781

**Published:** 2021-04-20

**Authors:** Carlos de Teresa, Alfonso Varela-López, Susana Rios-Álvarez, Rafael Gálvez, Coralie Maire, Santos Gracia-Villar, Maurizio Battino, José L. Quiles

**Affiliations:** 1Functional and Sports Medicine Service, Quirónsalud Hospital, 29004 Malaga, Spain; cdeteresa@teryos.com (C.d.T.); srios@teryos.com (S.R.-Á.); 2Biomedical Research Centre, Institute of Nutrition and Food Technology “José Mataix Verdú”, Department of Physiology, University of Granada, Avenida del Conocimiento s/n., 24560 Granada, Spain; alvarela@ugr.es; 3Pain Universitario Virgen de las Nieves University Hospital, 18012 Granada, Spain; rafaelgalvez@hotmail.com (R.G.); coraliem050@gmail.com (C.M.); 4Research Center for Foods, Nutritional Biochemistry and Health, Universidad Europea del Atlántico, 39011 Santander, Spain; santos.gracia@uneatlantico.es; 5Research Center for Foods, Nutritional Biochemistry and Health, Universidad Internacional Iberoamericana, Campeche 24560, Mexico; 6Department of Clinical Sicences, Università Politecnica delle Marche, 60131 Ancona, Italy; m.a.battino@univpm.it; 7International Research Center for Food Nutrition and Safety, Jiangsu University, Zhenjiang 212013, China; 8Research Group on Food, Nutritional Biochemistry and Health, Universidad Europea del Atlántico, 39011 Santander, Spain

**Keywords:** bioelectronics, electromagnetic signals, chronic low back pain, neuropathic pain, quality of life, functional capacity

## Abstract

Low energy pulsed electromagnetic signals (PEMS) therapy, in the field of bioelectronics, has been suggested as a promising analgesic therapy with special interest in treating conditions with poor response to pharmacotherapy. This study evaluated the effectiveness of PEMS therapy on the treatment of chronic low back pain patients with a neuropathic component. A group of 64 individuals with such condition was allocated to a 2-week treatment period (10 twenty-minute sessions on consecutive days) with an active PEMS therapy device or an inactive device in random order. The pain was assessed on a visual analog scale, and the functional status was assessed using the SF-12 questionnaire. The visual analog scale scores were lower after treatment than at baseline but only in the group treated with the active device. According to the DN4 score, neuropathic pain decreased in both experimental groups with respect to baseline, but this was only significant for the group treated with the active device. Similarly, an improvement in the SF-12 and Medical Outcomes Study (MOS) sleep scale components was reported. The study demonstrated that low-energy PEMS therapy was efficient in reducing pain and improving function in chronic low back pain patients with a neuropathic component.

## 1. Introduction

Low back pain is a health problem characterized by high prevalence in the general population, particularly in Western societies, with significant economic and social repercussions. Low back pain is one of the major reasons for medical treatment seeking and is one of the most challenging chronic pain disorders to treat [1]. In addition, low back pain has become one of the first causes of work absenteeism [2]. It is expected that between 60% and 80% of the world population will experience low back pain during their lifetime [3], with 65% being recurrent and longstanding episodes. It becomes chronic by up to 5–10%, it evolves to chronic disability and consumes up to 75% of the total resources dedicated to lumbar disease [4,5,6,7,8,9]. Although low back pain can be caused by different muscle-skeletal etiologies (muscle overcharge, ligament strains, herniated discs, osteoarthritis, scoliosis, or osteoporosis-related fractures), in most cases (80%), the cause of low back pain cannot be attributed to any specific structural lesion [4]. The pathophysiology of back pain is complex and, although mechanical and inflammatory responses are very frequent, peripheral and central neurological sensitization processes have a relevant role when the neuropathic component is predominant [10,11,12]. Estimates of the proportion of chronic low back pain patients with a neuropathic component range from 17 to 54% and vary according to the diagnostic method used [10,12,13,14,15]. However, the neuropathic component in chronic back pain may be underestimated [16] because approximately 28% of patients had an uncertain diagnosis [11]. 

Compared to inflammatory pain, neuropathic pain has a greater impact on sleep disturbances, anxiety, and reduction in quality of life and functional capacity [15,17,18,19], and from the economic point of view, a higher healthcare expenditure (high pharmaceutical cost, and increase in the number of emergencies, hospitalizations, and complications) [15,17,20]. Despite the negative economic impact, therapies for neuropathic pain show only modest evidence of efficacy. Typically, no more than half of patients experience clinically significant pain relief with currently available pharmacotherapy, and these drugs are associated with a risk of significant adverse effects. Despite the variety of treatments available, no modality or therapeutic approach has stood out as a definitive solution [21,22]. Thus, there is still a demand for new approaches, less invasive and free of side effects. Non-pharmacological options for the management of chronic low back pain may also include noninvasive approaches, such as transcutaneous electrical nerve stimulation (TENS) and invasive procedures, including epidural steroid injections (ESIs) and spinal cord stimulation (SCS) [23,24]. 

Within the field of bioelectronics, the use of electromagnetic signals has had a notable increase in the past decade in acute and chronic rehabilitation treatments, and it provides a noninvasive, safe, and accurate method to stimulate therapeutic targets depending on the tissue, functional or structural location, and the type or mechanism of pain or disease [25,26,27]. It has been reported that pulsed electromagnetic signals (PEMS) therapy yields several benefits into bone metabolism and osteogenesis, acute pain relief, wound healing, edema, and inflammation control, as well as in chronic pain associated with connective tissue injury and joint-associated soft tissue injury (such as in fibromyalgia and osteoarthritis) [28,29,30,31,32]. In addition to its positive health effects, PEMS therapy has other benefits such as enhancing its potential of compliance (due to being noninvasive, low-risk level, and its absence of side effects), selectivity to act on physiological and therapeutic targets [21], and positive results regardless of whether the low back pain is acute or chronic, or whether it has a muscular, discogenic, or complex origin [33,34,35]. However, the number of studies available in the scientific literature investigating the effectiveness of PEMS on low back pain is still scarce, and even fewer if we consider low back pain subgroups [36].

According to all the above mentioned, the present study aimed to evaluate the analgesic efficacy of PEMS therapy in patients with chronic non-specific low back pain associated with a neuropathic component using a noninvasive bioelectronic device (Physicalm^®^, Biotronic Advance Develop^®^, Granada, Spain), with a monopolar capacitive dielectric PEMS transmission. 

The primary outcome of the study is analyzing the effectiveness of the PEMS therapy in reducing pain intensity and the neuropathic component. In addition, the effects of treatment on the amount and quality of sleep and health-related quality of life have been analyzed.

## 2. Materials and Methods

### 2.1. Study Design

The research reported in this article consisted of a prospective, open-label, randomized and sham-controlled, double-blind pilot study of parallel groups, with repeated measures at pretreatment (baseline), 1 month post-treatment, and 3-month follow-up. This study was designed to assess the effect of PEMS therapy on pain and disability in patients with non-specific low back pain complaints. The trial was conducted at the Virgen de las Nieves Hospital, Granada, Spain. All procedures were in accordance with the local research ethics committee and the 1964 Helsinki Declaration and its later amendments or comparable ethical standards. The trial protocol was approved by the institutional ethics committee before recruitment, screening, and data collection. At the initial phase (Visit 0), the protocol was explained to the potential participants, and they were asked to sign the informed consent. Informed consent and clinical history were obtained from all individual participants included in the study.

#### 2.1.1. Study Participants

Patients diagnosed with non-specific chronic low back pain associated with a neuropathic component, referred to the Pain Treatment Unit of the Virgen de las Nieves Hospital in Granada (Spain), were recruited for study participation. Various inclusion and exclusion criteria (shown in Table 1) were used to minimize the confounders and allow for a stringent study design.

#### 2.1.2. Sample Size

There are few previous studies in the literature demonstrating statistically significant differences in the efficacy of the treatment. Given these conditions, the present research has been developed as a pilot study of a total of 60 patients; including the relevant cases of neuropathic pain that were considered interesting for the study. Such sample size has been determined considering the time of application of the sessions and the limited time available to complete the study (3 months).

#### 2.1.3. Randomization

Once the eligibility of the participants according to the inclusion and exclusion criteria was verified and they signed an informed consent to participate in the study, they were sequentially and randomly assigned to either the intervention group (treated with the active device) or the control group (treated with the inactive/sham device) at the time of enrollment. The assignment was performed using a computer-generated sequence guarded and created by a person belonging to the research support staff of the FIBAO (Andalusian Public Foundation for Biosanitary Research of Eastern Andalusia), who was outside the research group, through a computerized system of simple randomization 1:1 (M.A.S. 2.1 @ Glaxo Wellcome). The used software generates a sequence of numbers randomized and assigned to each patient (in order of inclusion into the study) to the corresponding group.

#### 2.1.4. Intervention

Patients allocated to the intervention group were locally exposed to a therapy based on the administration of low energy PEMS, via noninvasive transcutaneous and dielectric capacitive monopolar transmission to the target tissues, using a bioelectronic device (Physicalm^®^, Biotronic AD^®^, Granada, Spain) at the painful area in twenty-minute sessions every day (visit 1–10), except the weekends, for 2 weeks in active mode. In contrast, subjects assigned to the control group, underwent all procedures similarly to the treatment arm, except that the device remained in inactive mode during the session. Participants in the intervention group received PEMS therapy with a pulsed emission of 840 kHz low intensity and pulses of the order of nanoseconds, with a pulse/pause ratio of 500 μs and an average work cycle as applied energy of 45%, and the emissions were modulated and processed digitally. The PEMS therapy application was dynamic, with a continuous rotation and translational movement on the dorsolumbar spine, making focused energy deposits in the areas involved in the painful process. Five milliliters of almond oil were used to improve gliding along with the 20-min application of PEMS [37,38].

The treatment sessions were conducted at the Hospital Pain Unit. A research assistant telephoned the participants up to three times per week for the first week and then weekly after that to monitor for any changes or adverse effects and encourage compliance. Treatment compliance and medication use were assessed using daily self-reported diaries.

Patients allocated to the control group were treated with another device with the same characteristics in inactive mode (placebo or “sham”), i.e., without selecting any pain program. Therefore, no electromagnetic signal energy was transmitted when the device was turned on. Sham devices were externally identical to the active devices, and they were also connected to the current, allowing the pilots to light up, mimicking the intervention device. No vibration, heat, or other tactile stimuli was generated by any of these devices. Therefore, neither the patient nor the person managing the device was aware of whether the subject was receiving electromagnetic waves.

#### 2.1.5. Usual Care

All patients followed the usual pharmacological treatment for their painful condition (first and second step of the treatment guidelines: antiepileptics, antidepressants, weak opioids), according to the established guidelines that follow international recommendations (see Table 2). They did not undergo other types of treatments, such as physiotherapy or exercise, keeping only their basic medication. All patients were on sick leave when visiting the Pain Treatment Unit. Participants randomized to receive PEMS treatment were provided by a member of the study team. Patients taking strong opioids, topical pain reliever drugs, or NMDA antagonists must pre-wash at least 4 days before entering the trial. Moreover, to prevent pharmacotherapy from altering the results of the study, monitoring was carried out at the beginning of the study and during the different analysis points to ensure that the patients did not change the type of drug or dose during the entire experimental period.

#### 2.1.6. Safety Criteria

All subjects with at least one intervention of PEMS were included in the safety population. Patients were followed up throughout the study, and all local tissue effects and adverse events were recorded.

### 2.2. Pain Intensity Assessment

The intensity of self-reported pain was measured using a visual analog scale (VAS). The VAS is a single-item continuous scale comprising a 10-cm in-length horizontal line anchored by two verbal descriptors, one for each symptom extreme, “no pain at all” (score of 0) and “worst imaginable pain” (score of 10). The pain VAS is administered as a paper and pencil measure, and it is self-completed by the respondent and takes 1 min to complete. The respondents were asked to place a line perpendicular to the VAS line at the point that represents their pain intensity. Subjects rated their pain intensity in the VAS once a day for the five days preceding each visit in the morning upon rising and recorded it in a diary. Then distance in cm from the endpoint of the VAS labeled as “no pain at all” to the patient’s mark was measured with a ruler and the score of the subject’s pain was determined, so it was used as a numerical index of pain severity. Two VAS scores were calculated, the maximum VAS as the highest daily pain intensity measure collected during the last 5 days prior to each visit, and the baseline VAS, which was calculated as the average daily pain intensity measures collected during the last 5 days prior to each visit.

### 2.3. Secondary Outcome Measures

At baseline (visit 0), one month after finishing the treatment (visit 11, V11), and after a three-month follow-up (visit 12, V12), participants completed a battery of instruments to measure the biopsychosocial sequelae often associated with chronic low back pain. At visit 0, a clinical evaluation was performed in which the clinical anamnesis was conducted (clinical history and administration of health scales and questionnaires) along with protocol explanation. Then, one month after the patients finished the treatment sessions (visit 11), they underwent a medical examination, and records, surveys, and health scales and questionnaires were provided again. Administrative staff set up an appointment for the next medical assessment, in consultation after 3 months (visit 12) for a similar evaluation. Different medical professionals conducted the initial and final assessment of the treatment in consultation with patients.

#### 2.3.1. Neuropathic Pain Assessment

Neuropathic pain was assessed by using the DN4 questionnaire, a validated and clinician-administered questionnaire developed by the French Neuropathic Pain Group. The DN4 includes four questions consisting of sensory descriptors and signs related to bedside sensory examination and was deliberately reduced to a minimum number of simple and presumably discriminant items requiring yes or no responses, also simplifying its scoring. The DN4 was developed to differentiate neuropathic and nociceptive pain and consists of 7 items related to symptoms and 3 items related to the clinical examination [39]. Question I included 3 items related to the description of pain: burning squeezing, painful cold, electric shocks. Question II included 4 items related to the association of paresthesia/dysesthesia within the painful area: tingling, punctures, numbness, and stinging. Question III included 2 items related to sensory deficits: touch hypoesthesia and pricking hypoesthesia. Question IV included 1 item related to evoked pains: touch-evoked allodynia. Examination of sensitivity to touch and pricking was performed using a finger at a pressure that does not provoke pain in a normal area or a soft brush and a Von Frey hair (no. 13, Somedic), respectively. The soft brush (three movements) was also used to evaluate tactile (i.e., dynamic mechanical) allodynia. A score of 1 is given to each positive item and 0 to each negative item. The total score was defined as the sum of the responses to all the items. The cut-off value for diagnosing neuropathic pain is a total score of 4 more out of 10.

#### 2.3.2. Sleep Quality and Quantity Assessment

Quality and quantity of sleep were assessed using the Medical outcomes study (MOS) sleep scale [40], a subscale of the MOS health status measure. This questionnaire is a patient-reported, non-disease-specific tool for evaluating the sleep outcomes consisting of 12 items assessing the key constructs of sleep. It is self-administered and takes approximately 3–5 min to complete. Respondents are asked to recall sleep-related activities over the preceding four weeks. The MOS Sleep Scale measures subjective experiences of sleep across six different domains or subscales, and as such, may be potentially relevant for evaluating sleep problems. These domains are: (1) Sleep disturbance (have trouble falling asleep, how long to fall asleep, sleep was not quiet, awaken during your sleep time and have trouble falling asleep again), which measures the ability to fall asleep and maintain restful sleep; (2) sleep adequacy (get enough sleep to feel rested upon waking in the morning, get the amount of sleep needed), which measures the sufficiency of sleep in terms of sleeping enough to provide restoration of wakefulness; (3) sleep quantity, which measures (in hours) the amount of sleep an individual has had each night; (4) somnolence, which measures daytime drowsiness or sleepiness (drowsy during the day, have trouble staying awake during the day, take naps); (5) snoring; and (6) shortness of breath, or headache, snoring, awaken short of breath or with headache, and quantity of sleep. Ten out of the 12 scale items are scored using a six-point (All of the Time, Most of the Time, A Good Bit of the Time, Some of the Time, A Little of the Time, and None of the Time) response scale, one item uses a five-point Likert scale, and sleep quantity is an open-ended question recording the actual number of hours slept per night. In addition, a summary index, the sleep problems index II, a measure that assesses overall sleep problems, was constructed with 9 out of the 12 items, respectively, to provide the composite scores. This index contains questions from the sleep disturbance, sleep adequacy, respiratory impairment, and somnolence domains but does not include the question on sleep quantity. All domains except sleep quantity are recalibrated on a 0–100 scale representing the percentage of a particular sleep domain; sleep quantity is recorded as 0–24 h. Higher scores for the domains of sleep disturbance, somnolence, and the sleep indices indicate worse sleep problems. Lower scores for sleep quantity and sleep adequacy indicate worse sleep problems.

#### 2.3.3. Health-Related Life Quality Assessment

Health-related life quality was assessed using the version of the SF-12 Health Survey questionnaire [41,42] adapted for Spain by Alonso et al. [43,44]. The questionnaire was administered by an interviewer in a personal interview and it was completed in less than 2 min. This tool provides a profile of the state of health and is one of the most widely used generic scales in evaluating clinical results, being applicable both for the general population and for patients with a minimum age of 14 years and both in descriptive and clinical studies. The SF-12 consists of 12 items from the 8 dimensions of the SF-36: Physical Function (2), Social Function (1), Physical role (2), Emotional role (2), Mental health (2), Vitality (1), Body pain (1), General health (1). The response options form Likert-type scales that assess intensity or frequency, and the number of answer options ranges from three to six, depending on the item. It allows obtaining two summary scores: physical and mental summary measures. To facilitate interpretation, these scores were standardized with the values of the population norms so that 50 (standard deviation of 10) is the mean of the general population. For each of the 8 dimensions, the items are coded, aggregated, and transformed on a scale ranging from 0 (the worst health status for that dimension) to 100 (the best health status).

### 2.4. Statistical Analyses

Demographic and clinical characteristics of patients allocated at each group were described as mean and standard deviation for continuous variables and absolute and relative frequencies for categorical variables. Differences in continuous variables between the two experimental groups at any visit were analyzed by the independent group student’s t test for unpaired samples or Mann–Whitney test. Categorical variables were compared for differences between the two experimental groups using the χ2 test. The validity conditions of the test have been considered (number of cells with expected frequency less than 5 not higher than 55–30%). Comparisons between the values obtained at each visit and the values obtained at the baseline visit for each experimental group were performed. For these comparisons, the McNemar test for paired data for categorical variables with two levels and the two-factor generalized linear mixed model Manova analysis of repeated measures for ordinal variables were used. Means and standard errors were used as descriptive parameters for interaction factors. For each factor, the Mauchly sphericity test (equality of variances between the differences in the levels of the visit factor) and the Levene test (equality of variances between the levels of the device factor) have been performed. Where a significant result was disclosed, a post-hoc comparison between visits (for each device) and between devices (at each visit) with nonparametric test of Wilcoxon-Mann–Whitney was also performed. The Mauchly sphericity test and the Levene test have been performed in each of them to contrast the hypotheses of equality of variances between the differences in the levels of the visit factor and between the levels of the device factor, respectively. The Wilcoxon and Mann–Whitney nonparametric test was used to analyze the differences between visits for each treatment and between treatments at each visit. To address the issue of potential confounders, univariate analyses were performed to assess the potential effects of age, sex, and baseline pain VAS score on change in pain intensity. The χ2 analysis was used to assess the relationship between sex and change in pain severity. Pearson correlation analyses were performed to assess the relationships for age and pretreatment pain score versus change in pain severity. A statistically significant difference was considered when the significance level was less than 5% (*p* < 0.05). All statistical analyses were performed using the statistical software SPSS Statistics version 24.0 for Windows (SPSS Inc, Chicago, IL, USA; IBM Corp., Armonk, NY, USA).

## 3. Results

Seventy-seven subjects were randomly assigned to receive either PEMS (*n* = 39) or sham treatment (*n* = 38) (Figure 1). A total of 12 out of the 77 patients who started the treatments, dropped out during the study. Thus, 84% of the enrolled patients completed the study. Although the initial intention was to conduct an intent-to-treat analysis, a consensus decision was made not to analyze the 12 dropouts for the following reasons. Five patients dropped out of the study during the active treatment period. Seven patients did not attend all hospital sessions, so they undertake a discontinued either PEMS or sham treatment. Consequently, only 65 out of the 77 patients were included in the final analysis: 34 of 39 (87.2%) randomly assigned to receive PEMS (Device A), and 31 of 38 (81.6%) randomly assigned to receive sham treatment (Device B). No adverse events occurred during the active treatment period. Participants and drop-outs were demographically and clinically similar.

### 3.1. Sociodemographic Variables

The study was then performed finally with full data from 65 individuals, 34 in treatment group and 31 in control group (Figure 1). Sociodemographic variables are shown in Table 3. No differences in sex, civil status, or economic activity were found between the experimental groups. Variable distributions did not differ significantly between groups. No differences between the two experimental groups for gender distribution, age, studies, civil status, or economic activity were found.

### 3.2. Self-Reported Pain Intensity

Only the treated group had improved VAS pain scores at one and three months after finishing sessions (Figure 2 and Figure 3). In the treatment group, pain intensity decreased through visits. Basal VAS scores obtained at each visit from visit 2 were lower than those obtained at baseline (visit 0) in treated patients, whereas the values of this parameter in the control group were significantly different from baseline values at visit 1, 2, 6, and 7 (Figure 2). However, the difference not always represented an improvement, and in the case of it, this was significantly higher in the treatment group at the same visit than in the control group. Regarding maximum VAS scores, these were lower than those reported at baseline from visit 2, but no significant differences were found between baseline maximum VAS scores and those reported at any visit in the control group. However, differences between the two experimental groups were only statistically significant from visit 4 for both basal and maximum VAS scores with treated patients reporting lower values (Figure 3).

### 3.3. Neuropathic Pain Components

Average total scores obtained using the DN4 questionnaire are presented in Table 4, whereas frequencies of individuals with a score of 4 or more out of 10 (neuropathic pain diagnosis) are presented in Table 5. At baseline (visit 0), all patients have a total score of 4 or more, and the mean score did not differ between experimental groups. The number of individuals with a score of 4 or more after ending treatment sessions (visit 11 and visit 12) decreased in the treatment group compared to that at baseline. At visit 11, only 6 individuals had a score of 4 or more, whereas 25 patients in the control group had a score of 4 or more. Similarly, a higher number (26) of individuals presented a score of 4 or more in the control group than in the treatment group, although the number (8) of individuals with a score of 4 or more in the control group was a little higher than in the previous visit. Similar differences were found in mean DN4 scores that indicated an improvement at visits 11 and 12 compared to the baseline (visit 0), but only in the treatment group.

### 3.4. Sleep Quality and Quantity

The average scores obtained in the MOS–sleep questionaries are shown in Table 6. Treatment was associated with statistically significant improvement in sleep quality after ending all treatment sessions. At visits 11 and 12, scores for almost all sleep domains were higher than values obtained at visit 0, although such improvements were found for only three of the four items of the MOS–sleep disturbance subscale. Moreover, the MOS–sleep quantity subscale improvements were statistically significant for the treatment group compared to the control group. An exception was the time to fall asleep that was no different from the baseline in the treatment group, although it was increased at visits 11 and 12 in the control group. Still, the treatment group showed no improvement in the quantity of sleep. Improvement in snoring at the two endpoint visits (visits 11 and 12) was found in the control group; however, it was higher in the treatment group score. The average quantity of sleep reported by subjects in the control group was lower after ending treatment sessions, and it was lower than the treatment group.

### 3.5. Life Quality

At baseline, the two SF-12 components were similar in both experimental groups. PCS (physical component summary score) improved in treated individuals, but in the control group was even worse than at baseline. As expected, the treated group reported a higher score than the control group in subsequent visits. In contrast, no differences were found using the MCS (mental component summary score) (Table 7).

## 4. Discussion

The current randomized trial was designed to evaluate the clinical improvement in a cohort of patients with non-specific chronic low back pain associated with a neuropathic component after treatment with PEMS by using a bioelectronic therapeutic device (Physicalm^®^, Biotronic Advance Develops^®^, Granada, Spain), via noninvasive transcutaneous to transmit electromagnetic signals through a dielectric, capacitive monopolar transmission. Pain intensity describes how much a patient is in pain [45] and is a critical component of pain experience and pain severity. Thus, the successive measurement of the intensity of low-back pain under different conditions is a key parameter for understanding the evolution of the patient’s recovery process. Pain VAS is a reliable and valid parameter [46] that has been found to correlate positively with other self-reporting measures of pain intensity [47,48], and it has been demonstrated to be sensitive to treatment effect [49]. Pain VAS scores showed a reduction in pain intensity through treatment sessions in the experimental group. In addition, the difference in pain intensity measured using VAS at two different time points represents the real difference in magnitude of pain, which seems to be the major advantage of this PEMS therapy compared to other treatments. However, the clinical importance of pain intensity is not always easy to determine [50]. Several attempts have been made to identify the amount of change necessary to be clinically significant in pain VAS [49]. Analgesic interventions providing a change of 10 units out of 100 in the VAS represents a clinically meaningful improvement or deterioration, and a VAS of 33 or less supposes an acceptable control of the pain. However, this ratio is more reliable at the group level than at the individual level. A change of approximately 20% for chronic back pain and 12% for acute pain is considered clinically significant [51]. In the present study, the average difference in pain VAS score compared to the baseline was 77.1 (V11) and 66.8 (V12) in the treatment group, whereas it was 23.5 (V11) and 11.7 (V12) in the control group. These results are particularly important considering that they were obtained from visits 11 and 12, one and three months after the end of treatment, respectively, showing that the beneficial effect is still maintained in the medium term.

PEMS therapy has been investigated in many studies searching for its effects in alleviating acute and chronic pain in musculoskeletal disorders [52]. These conditions include low back pain with different etiologies, such as generalized low back pain [33]; acute non-specific low back pain [21]; discogenic lumbar radiculopathy [34], and chronic low back pain [53]. All the mentioned studies reported a decrease in pain intensity compared to the baseline, with a mean difference in pain intensity from baseline to the endpoint from 21 to 64 points out of 100 on the VAS [36]. Moreover, the studies on discogenic lumbar radiculopathy and lumbar myalgia showed a large effect size [34,35]. In addition, no differences in such improvement between the PEMS-treated group and the sham group were found in one of the studies [21]. The inconsistent results of the PEMS treatment effect found in the literature could be a consequence of the high heterogeneity between the PEMS therapy protocols of the different studies. Overall, it can be suggested that PEMS greatly reduces the pain intensity in low back patients when used alone, independently of the low back pain condition. However, some studies found no additional benefits [36,52] when added to other standard therapies such as standard physiotherapy [21] or analgesic therapy [33], probably due to the presence of a predominant neuropathic component in the patients studied. In our study, all patients followed the usual pharmacological treatment for their painful condition (first and second step of the treatment guidelines: antiepileptics, antidepressants, weak opioids), having had poor analgesic effects. The analgesic improvement in our study may be due to the novel technique used to transmit PEMS (dielectric, capacitive and monopolar). This type of transmission adds important advantages because of the selectivity of the energy deposits on the therapeutic targets and its lack of side effects. Dielectric capacitive PEMS transmission [37] allows energy to be transferred noninvasively to deeper tissues in a more controlled and accurate manner than conventional electrotherapy or thermotherapy systems [38]. This system is based on tissue dielectric charge capacity [54] to deeply transport high-frequency energy and a higher degree of focalization and density of energy applied while avoids warming the surface [38].

The neuropathic component of low back pain was assessed using the DN4 questionnaire, whose discriminant validity for identifying neuropathic components, particularly in low back pain patients, has been confirmed in previous studies [14,39]. According to DN4 scores, PEMS treatment led to a clear improvement in this condition compared with placebo, with a very reduced number of individuals showing DN4 scores enough to consider that they have neuropathic pain. This is consistent with the decrease in pain intensity found in both groups, but no previous studies have evaluated the effect of PEMS on this component in patients with different low back pain conditions.

Besides pain intensity, pain-related interference with daily or at-work activities (i.e., disability) is also an important pain severity component. The SF-12 instrument was used to evaluate the increase in the quality of life perceived by the patients. This SF-12 is subdivided into two separate health constructs: the physical component summary score (PCS) and the mental component summary score (MCS). As the SF-12 is a reduced version of the MOS, it does not contain sleep behavior items. The MOS–Sleep scale, a face valid index of sleep disturbance with adequate established reliability with objective sleep measures, was also administered to evaluate the effects on sleep quality and quantity.

Health-related quality of life instruments have been widely recommended as an outcome measurement for patients with low back pain [55,56], and, interestingly, their reliability and validity are well established. In contrast, many physical impairments measurements have been found to be lacking in reliability and validity [57]. In this context, generic instruments such as the SF-12 questionnaire are designed for broad use in various patient populations to broadly assess the concepts of health, disability, and quality of life. Although generic health status measures are often less responsive to changes in specific conditions than disease-specific instruments, they are important for broad comparisons of the relative impact of different conditions or treatments on the health of the population. In addition, both instruments, the MOS SF-36 and its short-form, the SF-12, have been widely used to evaluate low back pain [58]. It is worth highlighting the excellent validity of the SF-12 criteria compared to the SF-36, because the SF-12 can explain, for both components, more than 90% of the variability of the summary scores of the original version of the questionnaire. Regarding reliability, the estimates were very high for the SF-36, close to 0.9, and lower for the SF-12, although they exceeded the expected standard of 0.7 for group comparisons [59]. In contrast to the original dimensions of the SF-36, CSF and CSM of the SF-12 are standardized and that their interpretation is based on standards. In addition, summary components allow the number of statistical comparisons to be reduced and presenting a more symmetric distribution eliminating the ceiling and floor effects [43]. This offers the advantage of providing a direct interpretation of the general Spanish population scores, which has a mean of 50 and a SD of 10. To calculate summary components of the SF-12, we used the weights calculated for Spain by Vilagut et al. [43] whose values for the different dimensions are very similar to those obtained for the different language versions of the questionary [60]. In the present study, both groups showed average components at baseline that indicated a worse state of health than the general Spanish population. Treatment improved the state of health, but it continued to be worse than in the general population.

Regarding the evaluation of the recovery of function, the authors use specific scales depending on the musculoskeletal diseases. Several scoring systems are frequently used in the clinical environment to measure the disability related to the low back condition. Up to date, studies investigating the effect of PEMS therapies on back pain did not evaluate the health-related quality of life by using the SF-12 questionnaire. However, the SF-12 tool was developed as an abbreviated version of the SF-36, containing a subset of 12 identical questions embedded in the SF-36 eight constructs [41]. In this sense, Park et al. [35] reported significant improvements in SF-36 prior to and following treatment in lumbar myalgia patients and no significant difference between the PEMS-treated group and the group receiving sham treatment. For these reasons, it was assumed that there was no significant difference between the two groups following the treatment [35]. Other scales and indexes used to evaluate and quantify the recovery of participant’s function in studies assessing PEMS therapy effect on back pain are the EuroQol-5 Dimension (EQ-5D) [35], the Modified Oswestry Low Back Pain Disability Questionnaire (OSW) [34], the Korean version of Oswestry Disability Index (ODI) [21,35] and Modified Version Functional Activity Scale [21]. In general, previous studies have demonstrated the effectiveness of PEMS in reducing the disability related to low back pain [33,34,35,53]. However, a study reported no significant improvements in the PEMS-treated group when compared to the control group [21]. The study conducted by Omar et al. [34] was an exception, achieving a large effect size, a 42% mean reduction after daily applications of PEMS therapy for 3 weeks. However, some caution should be taken when considering this study since they used an adapted score.

Lastly, no significant differences in the demographic information and the physical examination provided before the trial, including gender, age, studies, civil status, or economic activity, were found between the two experimental groups. In addition, no significant differences between the two experimental groups in terms of the values of VAS, DN4, SF-12, MOS sleep scale score were found at baseline. Thus, the extent of pain and disability and the quality of life due to low back pain and the physical characteristics of the subjects in the PEMS-treated group and the sham group were similar at baseline. 

Importantly, neither group showed side or abnormal effects throughout the trial. Although PEMS is safe, its safety needs to be furtherly evaluated for its use in long-term treatment, because a 2-weeks treatment with a three-month follow-up shows only medium-term results. Additional investigations and clinical trial studies are needed.

Notwithstanding, it is worth noting some limitations that might have influenced the results of our study. VAS has some disadvantages. In particular, it seems to be more difficult to understand than other measurement methods and hence, more susceptible to misinterpretations or “zero-values.” However, this is particularly true in elderly patients and when respondents are given good instructions and the limitations are borne in mind, the VAS is still a valuable instrument to assess pain intensity and changes due to therapy. Likewise, SF-12 summary component scores could be an oversimplification of the information collected by the individual dimensions because they are calculated from the original 8 dimensions. However, summary components can be interpreted together with the profile provided by the 8 dimensions to avoid possible errors in interpreting the results. The use of SF-12 in conjunction with a disability scale has been recommendable because each tool measures unique aspects of disability and health-related quality of life. However, there is some overlap between both instruments, particularly concerning physical-function assessment. Regarding the MOS sleep scale, the lack of a cut-off score undermines its utility as a clinical tool. Nevertheless, the MOS scale may be more useful in studies where the focus is general health status assessment, but information on sleep parameters is of interest. This study also has several strengths: it is a randomized double-blind placebo-controlled treatment study; all the patients, outcome assessors, and the statistician were blinded to the group assignment, the sample size was large enough, and compliance was high.

## 5. Conclusions

In the present study, PEMS therapy decreased the pain intensity in patients with chronic low-back pain with a neuropathic component. The results of this study may provide important health and economic benefits because of the high prevalence and the low therapeutic control of this condition. The major benefit in pain and disability control found in this study, compared with others performed with PEMS treatment, is mainly based on the use of a bioelectronic device that allows a dielectric, capacitive, and monopolar transmission, ensuring the bioavailability of the signal with therapeutic value in the target tissues, in the absence of side effects. 

Although the most effective pattern and mode of application of PEMS remain controversial, its application according to the new forms of noninvasive transmission offered by medical bioelectronics represents an important advance, and this type of technique can be postulated as one of the first-choice therapies in the management of low back pain with a neuropathic component. Despite the promising results obtained in this study, we should bear in mind and consider that this is a pilot trial. New studies with a larger number of patients and with more extended follow-up periods are needed to optimize the protocols for this type of treatment with PEMS and to corroborate the results obtained in the present research.

## Figures and Tables

**Figure 1 jcm-10-01781-f001:**
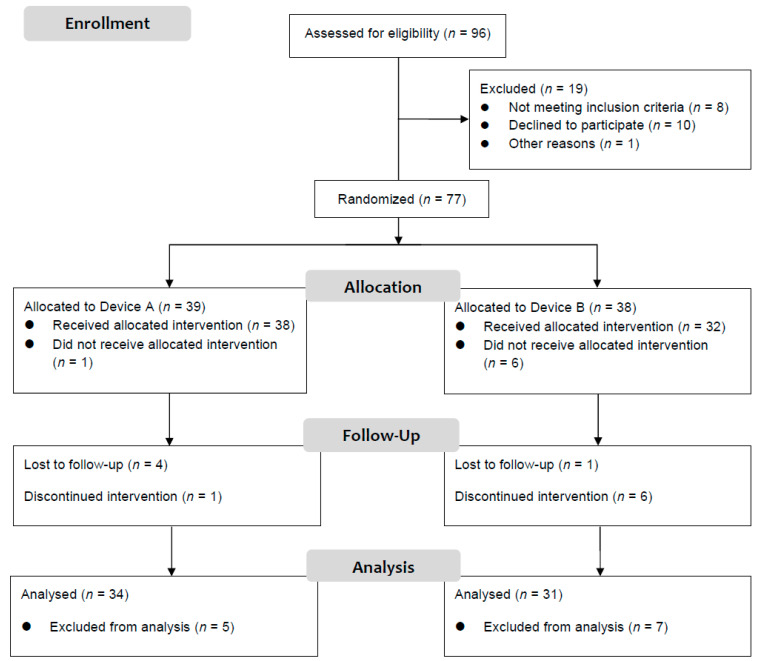
Study completion flowchart.

**Figure 2 jcm-10-01781-f002:**
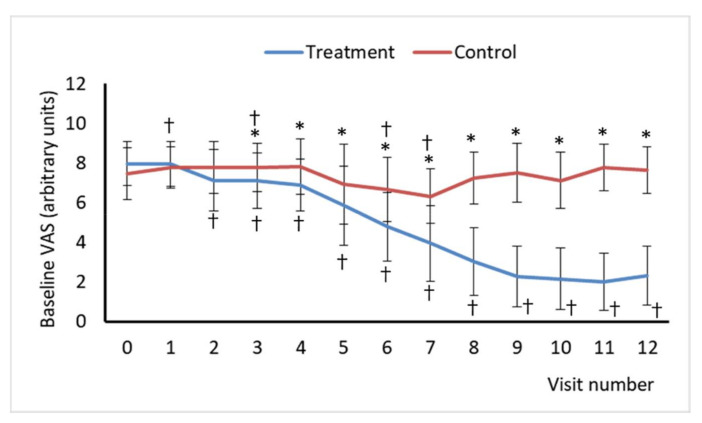
Baseline Visual Analogue Scale (VAS) of Pain scores for visits 0 to12. Baseline VAS was calculated as the average daily pain intensity measures collected during the last 5 days prior to the visit. Asterisk (*) indicates a statistically significant difference between the treatment and control groups at the same visit. Cross (†) indicates a statistically significant difference compared to the baseline (visit 0) for an experimental group.

**Figure 3 jcm-10-01781-f003:**
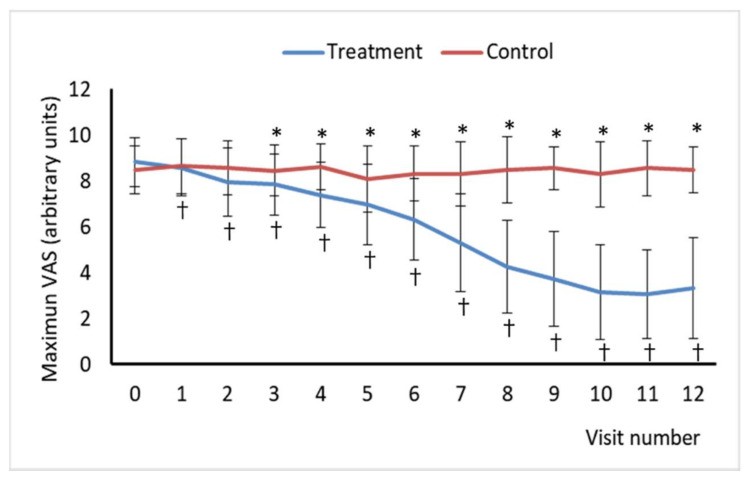
Maximum Visual Analogue Scale (VAS) of Pain scores for visits 0 to12. Maximum VAS represents the highest daily pain intensity measure collected during the last 5 days prior to the visit. Asterisk (*) indicates a statistically significant difference between the treatment and control groups at the same visit. Cross (†) indicates a statistically significant difference compared to the baseline (visit 0) for an experimental group.

**Table 1 jcm-10-01781-t001:** Inclusion and exclusion criteria.

Inclusion Criteria	Exclusion Criteria
Age > 18 years old Patients who have signed a written informed consent Patients diagnosed with neuropathic pain by using a DN4 questionnaire (DN4 score ≥ 4) Visual analog scale (VAS) pain score >3 at baseline Patients with pharmacological treatment from the first or second line of treatment (certain antiepileptic, antidepressant and/or weak opioids)	Patients pending assessment of disability, with suspicion of simulation or any serious psychiatric illness Participating in other studies simultaneously Patients with pacemakers or electrical or electronic system implanted Women who are pregnant or breast-feeding Patients with areas with undiagnosed skin conditions Presence of other painful concomitant diseases Patients with central neuropathic pain or general neuropathies Patients with other only somatic or visceral nociceptive pain Patients with inability to verbalize and discriminate data on intensity and location of pain Patients with active malignant neoplasms Patients taking strong opioids, topical pain reliever drugs or NMDA antagonists

**Table 2 jcm-10-01781-t002:** Allowed and prohibited medications during the trial.

Prohibited	Strong Opioids Topical Pain Reliever Drugs NMDA Antagonists
Allowed	Gabapentin and Pregabalin Duloxetine and Venlafaxine Tricyclic antidepressants NSAIDs and paracetamol or metamizole Opioids Minors: tramadol and codeine
Authorized rescue medication	Paracetamol (1 g every 8 h per day)

**Table 3 jcm-10-01781-t003:** Subject sociodemographic characteristics at baseline.

	Treatment		Control	
Gender	*n*	%	*n*	%
Male	13	38.2	14	45.2
Female	21	61.8	17	54.8
Age (mean (SD))	49.3	13.6	55.4	19.6
Studies				
No studies	2	5.9	3	9.7
Primary education	11	32.4	11	35.5
Secondary education	14	41.2	11	35.5
University education	7	20.6	6	19.4
Civil status				
Single	5	14.7	4	12.9
Married	20	58.8	17	54.8
Widower	1	2.9	6	19.4
Separated	8	23.5	3	9.7
Divorced	0	0	1	3.2
Economic activity				
Working	14	41.2	11	35.5
Unemployed	6	17.6	6	19.4
Retired	8	23.5	11	35.5
Housework	6	17.6	3	9.7

*n*, absolute frequency %, relative frequency; SD, standard deviation.

**Table 4 jcm-10-01781-t004:** DN4 questionnaire total scores.

	Treatment			Control		
Visit	*n*	Mean	SD	*n*	Mean	SD
V0	34	5.71 ^a^	0.21	31	5.13 ^a^	0.19
V11	34	2.53 ^b,^*	1.08	31	5.29 ^a^	1.55
V12	34	2.79 ^b,^*	1.04	31	5.26 ^a^	1.53

*n*, sample size for each experimental group; SD, standard deviation. V0, visit 0; V11, visit 11; V12, visit 12. For each visit, superscript letters, when are different between groups, indicates a statistically significant difference (*p* < 0.05) between experimental group at the same visit. Asterisk (*) indicates a statistically significant difference (*p* < 0.05) than baseline (vistit 0) for an experimental group.

**Table 5 jcm-10-01781-t005:** Patients with neuropathic pain diagnosis according DN4 questionnaire total scores.

	Treatment			Control		
Visit	*N*	*n*	%	*N*	*n*	%
V0	34	34 ^a^	100	31	31 ^a^	100
V11	34	6 ^b,^*	17	31	26 ^a^	81
V12	34	8 ^b,^*	24	31	27 ^a^	84

*N*, sample size for each experimental group; *n*, absolute frequency %, relative frequency. V0, visit 0; V11, visit 11; V12, visit 12. For each visit, letters, when are different between groups, indicates a statistically significant difference (*p* < 0.05) between experimental group at the same visit. Asterisk (*) indicates a statistically significant difference (*p* < 0.05) than baseline (vistit 0) for an experimental group.

**Table 6 jcm-10-01781-t006:** MOS–sleep scale questionaries results.

	Treatment					Control				
	*n*	Mean	SD	95%CI		*n*	Mean	SD	95%CI	
Sleep Disturbance										
Trouble falling asleep										
V0	34	3.53 ^a^	0.23	3.07	3.99	31	3.81 ^a^	0.24	3.33	4.29
V11	34	5.15 ^a,^*	0.18	4.79	5.51	31	3.74 ^b,^*	0.19	3.37	4.12
V12	34	5.06 ^a,^*	0.20	4.66	5.46	31	3.87 ^b^	0.21	3.45	4.30
Time to fall asleep										
V0	34	2.91 ^a^	0.24	2.44	3.38	31	2.613 ^a^	0.25	2.12	3.10
V11	34	2.88 ^a^	0.23	2.42	3.35	31	3.65 ^b,^*	0.24	3.16	4.13
V12	34	2.94 ^a^	0.23	2.49	3.39	31	3.81 ^b,^*	0.24	3.34	4.28
Restless sleep										
V0	34	3.88 ^a^	0.23	3.42	4.34	31	4.10 ^a^	0.24	3.61	4.58
V11	34	5.35 ^a,^*	0.20	4.95	5.76	31	3.87 ^b^	0.21	3.44	4.30
V12	34	5.27 ^a,^*	0.19	4.89	5.64	31	4.13 ^b^	0.20	3.74	4.52
Awaken during sleep										
V0	34	3.59 ^a^	0.17	3.24	3.934	31	3.48 ^a^	0.18	3.12	3.85
V11	34	4.77 ^a,^*	0.23	4.30	5.227	31	4.10 ^b^	0.24	3.61	4.58
V12	34	4.88 ^a,^*	0.23	4.43	5.337	31	2.87 ^b^	0.24	2.40	3.35
Snoring										
V0	34	3.47 ^a^	0.22	3.03	3.92	31	3.39 ^a,^*	0.23	2.92	3.85
V11	34	5.24 ^a,^*	0.21	4.81	5.66	31	4.03 ^b,^*	0.22	3.59	4.48
V12	34	5.06 ^a,^*	0.23	4.59	5.52	31	4.16 ^b,^*	0.24	3.67	4.65
Awaken Short of Breath or with Headache										
V0	34	3.53 ^a,^*	0.18	3.16	3.90	31	3.52 ^a^	0.19	3.13	3.90
V11	34	5.00 ^a,^*	0.27	4.47	5.53	31	4.52 ^b^	0.28	3.96	5.08
V12	34	4.82 ^a^*	0.23	4.36	5.29	31	3.48 ^b^	0.24	3.00	3.97
Quantity of Sleep										
V0	34	5.97 ^a^	0.22	5.54	6.40	31	6.45 ^a^	0.23	6.00	6.91
V11	34	6.09 ^a^	0.26	5.58	6.60	31	5.90 ^a,^*	0.27	5.37	6.44
V12	34	5.88 ^a^	0.24	5.39	6.37	31	5.29 ^a^*	0.26	4.78	5.80
Sleep Adequacy										
Enough sleep to feel rested										
V0	34	3.35 ^a^	0.21	2.94	3.76	31	4.03 ^a^	0.22	3.60	4.46
V11	34	5.20 ^a,^*	0.19	4.82	5.59	31	3.84 ^b^	0.20	3.44	4.24
V12	34	5.24 ^a^*	0.20	4.85	5.63	31	3.58 ^b^	0.20	3.17	3.99
Amount of sleep needed										
V0	34	3.65 ^a^	0.19	3.27	4.03	31	4.23 ^b^	0.19	3.83	4.62
V11	34	5.12 ^a^	0.23	4.66	5.58	31	3.90 ^b^	0.24	3.42	4.38
V12	34	5.27 ^a,^*	0.22	4.82	5.71	31	2.97 ^b,^*	0.24	2.49	3.44
Trouble staying awake during day										
V0	34	3.59 ^a^	0.17	3.24	3.93	31	3.48 ^a^	0.18	3.12	3.85
V11	34	4.77 ^a,^*	0.23	4.30	5.23	31	4.10 ^a^	0.24	3.61	4.58
V12	34	4.88 ^a,^*	0.23	4.48	5.34	31	2.88 ^b,^*	0.24	2.40	3.35
Take naps										
V0	34	3.53 ^a^	0.18	3.162	3.90	31	3.516 ^a^	0.19	3.13	3.90
V11	34	5.00 ^a,^*	0.27	4.468	5.53	31	4.516 ^a,^*	0.28	3.96	5.07
V12	34	4.82 ^a,^*	0.23	4.361	5.298	31	3.484 ^b^	0.24	3.00	3.97
Feel drowsy during day										
V0	34	3.44 ^a^	0.2	3.05	3.84	31	3.65 ^a^	0.21	3.23	4.06
V11	34	5.35 ^a,^*	0.17	5.02	5.68	31	4.03 ^b,^*	0.17	3.69	4.38
V12	34	5.18 *	0.2	4.75	5.60	31	3.81 ^b^	0.22	3.36	4.25

Abbreviations: *n*, sample size for each experimental group; SD, standard deviation; 95%CI: 95% confidence interval. V0, visit 0; V11, visit 11; V12, visit 12. For each parameter and visit, letters, when are different between groups, indicates a statistically significant difference (*p* < 0.05) between experimental group at same visit. Asterisk (*) indicates a statistically significant difference (*p* < 0.05) than baseline (vistit 0) for an experimental group.

**Table 7 jcm-10-01781-t007:** Questionnaire SF-12 on the State of Health.

	Treatment			Control		
Component	*n*	Mean	SD	*n*	Mean	SD
PCS-12						
V0	34	30.33 ^a^	8.09	31	31.63 ^a^	8.19
V11	34	34.88 ^a,^*	10.43	31	25.08 ^b,^*	6.82
V12	34	32.30 ^a,^*	11.01	31	30.31 ^a^	9.43
MCS-12						
V0	34	39.96	7.54	31	41.38	6.82
V11	34	39.15	7.40	31	44.91	7.44
V12	34	40.12	6.89	31	44.05	8.06

Abbreviations: *n*, sample size for each experimental group; SD, standard deviation. V0, visit 0; V11, visit 11; V12, visit 12. PCS, Physical Component Summary; MCS, Mental Component Summary. For each parameter and visit, letters, when are different between groups, indicates a statistically significant difference (*p* < 0.05) between experimental group at same visit. Asterisk (*) indicates a statistically significant difference (*p* < 0.05) than baseline (vistit 0) for an experimental group.

## Data Availability

The data presented in this study are available on request from the corresponding author. The data are not publicly available due to restrictions regarding patients privacy.

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
