# Peer review of "Evaluation of the Analgesic Efficacy of a Bioelectronic Device in Non-Specific Chronic Low Back Pain with Neuropathic Component. A Randomized Trial"

_jcm, 2021, doi:10.3390/jcm10081781_

Round 1

Reviewer 1 Report

Thank you for the opportunity to review manuscript entitled “Evaluation of the analgesic efficacy of a bioelectronic device in non-specific chronic low back pain with neuropathic component. A randomized clinical trial".

  • please provide registration number and name of trial registry. According to the CONSORT guidelines, all clinical trials should be registered.
  • was the sample size calculated on the basis of the primary outcome (pain intensity)?
  • please describe the interventions in detail, patients participated in 10-minute sessions daily for 2 weeks?
  • please describe in detail the phases of the study, when were the next measurements taken?
  • the flow chart is unreadable to me.
  • has the pharmacotherapy used not disturbed the obtained results?
  • were the patients working during the intervention or were they on sick leave?
  • did the patients perform any exercises, physiotherapy, or any other treatments?
  • "In the present study, PEMS therapy reduced pain intensity and disability in patients with chronic low-back pain with a neuropathic component." disability assessment tools were not used in the study. Please correct the conclusion.

Author Response

Evaluation of the analgesic efficacy of a bioelectronic device in non-specific chronic low back pain with neuropathic component. A randomized clinical trial.

Summary

The aim of this study was to evaluate the effectiveness of Pulsed Electromagnetic Signals therapy on the treatment of chronic low back pain patients with a neuropathic component. 64 individuals were allocated to an active PEMS therapy device or with an inactive device. The authors concluded that low-energy PEMS therapy was efficient in reducing pain and improving function in chronic low back pain patients with a neuropathic component.

 Main comments

I would recommend you have the manuscript checked by a native English speaker

*Done

The authors mention in the methods section that this study is a pilot study. However, this is not being followed up on. Could the authors clarify this?

*Follow-up was carried out at 1 month and 3 months after finishing the treatment, corresponding to Visits V11 (1 month after) and V12 (3 months after)

 Table 1 states that patients were diagnosed according by using a DN4 questionnaire. Can the authors clarify how the patients were clarified? What was the threshold for being included?

*As it was described in material and method section for DN4 questionnaire “the cut-off value for the diagnosis of neuropathic pain is a total score of 4 out of 10 or more. This has been clarified also in the Table.

Can the authors expand on the exclusion criteria: “patients taking prohibited medication”? What does this involve? This is being explained in Table 2, but it would be clearer if this was all mentioned together.

*The sentence “patients taking prohibited medication” has been modified to be clearer as follows: “patients taking strong opioids, topical pain reliever drugs or NMDA antagonists”

Even though the authors performed a randomized controlled trial, there might be an imbalance between possible important confounding factors between the treatment and control group. Did you control for confounding? It seems there was no controlling for confounding, and this might

* We agree with the reviewer that randomization is the most powerful tool to prevent confusion by homogeneously distributing variables between groups, both known and unknown. However, it should never be considered an absolute guarantee of absence of confusion. Notwithstanding, a variable must be associated with the exposure to become a confounding variable (Rothman & Greenland, 1998). In the present study, as indicated in the discussion, there was no significant differences between the two experimental groups in the demographic variables and in the physical examination given prior to the trial, including gender, age, studies, civil status or the economic activity as well as in terms of the values of VAS, DN4, SF-12, MOS sleep scale score at baseline. Thus, all known variables measured in the present study are homogeneously distributed between PEMS-treated group and the sham group, so they are not associated with exposure (i.e. treatment), and consequently they do not meet all the conditions to be a confounding variable. The authors appreciate the reviewer's observation because it has allowed us to delve into this question and eliminate a fragment of the discussion where it was mentioned as a possible weakness of the study.

Rothman KJ, Greenland S. Modern Epidemiology. Washington: Lippincott-Raven, 1998

Please explain the difference between figure 2 and figure 3. Are these chance scores comparing to baseline (figure 2), and comparing to the maximum value (figure 3). This is unclear.

* Maximum pain intensity was the highest daily pain intensity measure collected during the last 5 days prior to the visit and baseline pain intensity was calculated as the average of daily pain intensity measures collected during the last 5 days prior to the visit. This has been clarified in Figures and Material and methods section.

Why are only visits 9, 11 and 12 reported for the secondary outcomes? All other outcomes should at least be reported in the appendix.

*As it is indicated in material and methods: “At baseline (Visit 0), one month after finishing treatment (visit 11), and after a three-month follow-up (visit 12), participants completed a battery of instruments to measure the biopsychosocial sequelae often associated with chronic low back pain”. Therefore, only visits 0, 11 and 12 reported for the secondary outcomes. This has been written in the text.

The results are promising, but the discussion should be more nuanced, considering that the authors present this study as a pilot study.

* The authors greatly thank the reviewer for his comments. In effect, it is a pilot test and therefore the results must be considered in this context. In any case, the results are what they are and should not be underestimated. However, and in accordance with the concept expressed by the reviewer, a sentence has been included in the conclusions in this regard: "Despite the promising results obtained in this study, it should not be forgotten that it is a pilot trial, and as such it must be taken into account. New studies with a greater number of patients and with longer follow-up periods are needed to optimize the protocols for this type of treatment with PEMS and to corroborate the results obtained in the present research".

The discussion is unnecessary long, and I recommend shortening this section of the manuscript.

*Discussion has been reviewed and modified

Minor comments

 The flowchart misses the number of completers on the 32 weeks assessment for the Device B

*The flowchart has been modified

Line 366 is not phrased properly. What do the authors mean with “….was a little higher respect than …..”

*The sentence has been modified

Can the authors check that all abbreviations are fully written out the first time they’re being used.

*This issue has been reviewed.

Reviewer 2 Report

Peer review

Evaluation of the analgesic efficacy of a bioelectronic device in non-specific chronic low back pain with neuropathic component. A randomized clinical trial.

Summary

The aim of this study was to evaluate the effectiveness of Pulsed Electromagnetic Signals therapy on the treatment of chronic low back pain patients with a neuropathic component. 64 individuals were allocated to an active PEMS therapy device or with an inactive device. The authors concluded that low-energy PEMS therapy was efficient in reducing pain and improving function in chronic low back pain patients with a neuropathic component.

Main comments

  • I would recommend you to have the manuscript checked by a native English speaker
  • The authors mention in the methods section that this study is a pilot study. However, this is not being followed up on. Could the authors clarify this?
  • Table 1 states that patients were diagnosed according by using a DN4 questionnaire. Can the authors clarify how the patients were clarified? What was the threshold for being included?
  • Can the authors expand on the exclusion criteria: “patients taking prohibited medication”. What does this involve? This is being explained in Table 2, but it would be clearer if this was all mentioned together.
  • Even though the authors performed a randomised controlled trial, there might be an imbalance between possible important confounding factors between the treatment and control group. Did you control for confounding? It seems there was no controlling for confounding, and this might
  • Please explain the difference between figure 2 and figure 3. Are these chance scores comparing to baseline (figure 2), and comparing to the maximum value (figure 3). This is unclear.
  • Why are only visits 9, 11 and 12 reported for the secondary outcomes? All other outcomes should at least be reported in the appendix.
  • The results are promising, but the discussion should be more nuanced, considering that the authors present this study as a pilot study.
  • The discussion is unnecessary long, and I recommend shortening this section of the manuscript.

Minor comments

  • The flowchart misses the number of completers on the 32 weeks assessment for the Device B
  • Line 366 is not phrased properly. What do the authors mean with “….was a little higher respect than …..”
  • Can the authors check that all abbreviations are fully written out the first time they’re being used.

Author Response

Thank you for the opportunity to review manuscript entitled “Evaluation of the analgesic efficacy of a bioelectronic device in non-specific chronic low back pain with neuropathic component. A randomized clinical trial".

Please provide registration number and name of trial registry. According to the CONSORT guidelines, all clinical trials should be registered.

*This is not a clinical trial, it is only a study in humans, the manuscript has been reviewed to remove this term

Was the sample size calculated on the basis of the primary outcome (pain intensity)?

*No, as it was explained in section 2.1.2 Sample size “there are few previous studies in the bibliography demonstrating statistically significant differences in the efficacy of the treatment. Given these conditions, it has been developed as a pilot study of a total of 60 patients; including the relevant cases of neuropathic pain that were considered interesting for the study. Such sample size has been determined considering the time of application of the sessions and the limited time available for completion of the study (3 months).”  To avoid misunderstanding, section has been modified to remove information about theorical models to use if there was available data in bibliography.

Please describe the interventions in detail, patients participated in 10-minute sessions daily for 2 weeks?

* This has been clarified as follows: “at painful area in twenty-minutes sessions for a 2-week period in active mode all days except the weekends (visit 1-10)”

Please describe in detail the phases of the study, when were the next measurements taken?

*Section 2.3 Secondary outcome measures has been modified to clarify this issue.

The flow chart is unreadable to me.

*The flow-chart has been modified to facilitate understanding.

Has the pharmacotherapy used not disturbed the obtained results?

* These aspects have been detailed and clarified in section 2.1.5 on Usual care. To prevent pharmacotherapy from altering the results of the study, monitoring was carried out at the beginning of the study and during the different analysis points to ensure that the patients did not change the type of drug or dose during the entire experimental period. This sentence has been added to the above mentioned section. 

Were the patients working during the intervention or were they on sick leave?

*All patients were on sick leave when visiting the Pain Treatment Unit. This information is now provided in the subsection 2.1.5 Usual Care

Did the patients perform any exercises, physiotherapy, or any other treatments?

*They did not undergo other types of treatments, neither physiotherapy nor exercise, keeping only their basic medication. This information is now provided in the subsection 2.1.5 Usual Care

"In the present study, PEMS therapy reduced pain intensity and disability in patients with chronic low-back pain with a neuropathic component." disability assessment tools were not used in the study. Please correct the conclusion.

*It has been corrected

Round 2

Reviewer 1 Report

Thank you for the opportunity to review manuscript entitled “Evaluation of the analgesic efficacy of a bioelectronic device in non-specific chronic low back pain with neuropathic component. A randomized trial".

This revised manuscript has been much improved and is in a nice condition now.

Reviewer 2 Report

The authors have sufficiently processed the feedback.